# Evaluation of the Obstetric Patient: Pregnancy Outcomes during COVID-19 Pandemic—A Single-Center Retrospective Study in Romania

Melinda Ildiko Mitranovici [1,*] , Diana Maria Chiorean [2], Ioan Emilian Oală [1], Izabella Petre [3] and Ovidiu Simion Cotoi [2,4]

1 Department of Obstetrics and Gynecology, Emergency County Hospital Hunedoara, 14 Victoriei Street, 331057 Hunedoara, Romania; oalaioanemilian@gmail.com
2 Department of Pathology, County Clinical Hospital of Targu Mures, 540072 Targu Mures, Romania; chioreandianamaria@yahoo.com (D.M.C.); ovidiucotoi@yahoo.com (O.S.C.)
3 Department of Obstetrics and Gynecology, Victor Babes University of Medicine and Pharmacy, 2 Eftimie Murgu Sq, 300041 Timisoara, Romania; petre.izabella@umft.ro
4 Department of Pathophysiology, George Emil Palade University of Medicine, Pharmacy, Science and Technology of Targu Mures, 38 Gheorghe Marinescu Street, 540142 Targu Mures, Romania
* Correspondence: mitranovicimelinda@yahoo.ro

**Abstract:** Acute respiratory-syndrome-related coronavirus 2, or SARS-CoV-2, has become a public health issue in our country. It mainly affects the vulnerable population, especially those with comorbidities. In this retrospective study, we set out to explore the effects of COVID-19 on pregnancy, with the vulnerability of pregnant women to SARS-CoV-2 infection also representing a main focus. We included 39 patients who tested positive for SARS-CoV-2 and 39 control subjects recruited from the Emergency County Hospital of Hunedoara, Romania. Our aim was to explore the indirect impact of the COVID-19 pandemic on pregnancy, as our patient group was included in the "high-risk" category. As a result, cesarean section prevailed, the main reason being fetal hypoxia. Newborns were evaluated by real-time postnatal polymerase chain reaction (RT-PCR) viral testing: none exhibited SARS-CoV-2 infection, with no vertical transmission of the virus being detected. Moreover, we observed no maternal or neonatal deaths resulting from COVID-19. SARS-CoV-2 has been found to cause a heterogeneity of manifestations with damage to multiple organs, and its evolution remains unknown. In our study, the need for antiviral treatment was limited, but anticoagulants proved effective in terms of improving the outcome.

**Keywords:** COVID-19; maternal health; newborn health; maternal–child transmission

## 1. Introduction

In the recent years, we have experienced an extremely aggressive infection caused by a new coronavirus strain: the SARS-CoV-2 virus (severe acute respiratory syndrome coronavirus 2), whose main clinical manifestation is severe acute respiratory syndrome. The mode of transmission was found to be predominantly respiratory, through infected secretions or direct contact. Moreover, in Romania, it has led to a major public health issue. Impaired cell-mediated immunity in pregnancy turns pregnant patients into a vulnerable population. Anatomical and physiological changes during pregnancy, such as ascension of the diaphragm, changes in lung volume, vasodilation, mucosal edema, and anemia, all result in a tendency to hypoxia. In summary, the vulnerability of pregnant women to SARS-CoV-2 infection was shown to be increased [1], which was one of the main conclusions of our study.

According to the literature, the SARS-CoV-2 infection affects the outcome of the pregnant population, resulting in the appearance of severe pneumonia, acute respiratory distress syndrome (ARDS), multiorgan failure, or disseminated intravascular coagulation

(DIC), similar to that in the general population, with the occurrence of serious obstetric consequences, e.g., abortions, premature births, intrauterine growth restriction (IUGR), preeclampsia (PE), fetal hypoxia, and maternal and fetal or neonatal deaths [2]. As a consequence, an increase in the rate of cesarean sections was reported [3,4]. In contrast, there was no evidence of vertical transmission of this virus, either in the amniotic fluid, according to real-time postnatal polymerase chain reaction (RT-PCR) viral testing [3,5], or via breast milk, which is why breastfeeding was encouraged if the mother's condition allowed [6,7].

In terms of non-obstetric manifestations, various comparative studies that focus on populations of pregnant and nonpregnant patients have shown similarities [5,8,9].

Novoa, R.H. et al. describe a lethality rate among the general population of around 3–4%, pregnant women being classified as a high-risk population that requires intensive care in a proportion of 50% [10]. They are classified as an increased risk group, due to the anatomical and physiological changes during pregnancy, such as: reduced functional residual volumes, diaphragm elevation, altered cell immunity, increased thoracic transverse diameter, changes in pulmonary volume, vasodilation, and mucosal edema, which are associated with the tendency of pregnant women to hypoxia. Hormonal changes also play an important role in these vascular and respiratory modifications and also in pregnancy-specific procoagulant events [1,2]. The severity of the disease is associated with various risk factors, such as age, comorbidities, obesity, the healthcare system of their country or region, and their socio-economic status. However, it should be noted that data from the literature are based on a small series of studies [10–12].

The specific COVID-19 symptoms that are common in nonpregnant patients include coughing, fever, tachypnea, myalgia, dyspnea, sore throat, chest pain, nasal congestion, diarrhea, and nausea. The main changes observed in the laboratory tests were lymphopenia, leukopenia, anemia, increased polymerase chain reaction (PCR), thrombocytopenia, altered ferritin, increased levels of aspartate aminotransferase (AST) and alanine transaminase (ALT), increased levels of lactate dehydrogenase (LDH), increased cytolysis, and specific chest radiography changes [10].

It was interesting to observe how the SARS-CoV-2 infection affected pregnancy and fetal development. According to the literature, the maternal outcomes caused by SARS-CoV-2 virus infection include abortions, premature births, fetal distress (fetal asphyxia), stillbirth, preeclampsia (PE), diabetes mellitus (DM), maternal or neonatal death, and an increase in cesarean section [1,10]. Among the outcomes identified in newborns, the following were also observed: a low APGAR score, low birth weight, prematurity, intrauterine growth restriction (IUGR), persistent hypoxia, and rare vertical transmission of SARS-CoV-2 infection, all with possible recovery [10]. According to Allotey et al., some newborns can require intensive care services [12].

As regards COVID-19 and pregnant women, there are few serious case reports. As we are limited for ethical reasons, the study of SARS-CoV-2 virus infection during pregnancy remains a challenge [10].

Gabrieli et al. described another consequence of SARS-CoV-2 infection: the occurrence of thromboembolic events, which have yet to be elucidated. As pregnancy is a state of hypercoagulation, which is likely an adaptive mechanism to reduce the risk of hemorrhage during and after the delivery process, fibrinolytic activity is decreased and the appearance of venous stasis is common. In addition, the prolonged sedentary lifestyle caused by the lockdown and limitations of movement caused by cesarean section are also predisposing factors for thrombosis, thromboembolism, and thromboembolic risk factors in COVID-19 [13].

Several studies, including those conducted by Zitiello et al. and Novoa et al., refer to thrombocytopenia, which is why it was included as a biomarker for risk stratification [10,14].

As a result of the fatal thromboembolism that can occur as a consequence of infection with SARS-CoV-2, the pathophysiological mechanism was explored, with a resemblance

to TTP (thrombotic thrombocytopenic purpura) and changes in the proteolytic cleavage of von Willebrand factor–ADAMTS 13 (a disintegrin and metalloproteinase with a thrombospondin type 1 motif member 13) being noted [15,16].

The mechanism of DIC (disseminated intravascular coagulation) has also been studied in SARS-CoV-2 infection, which is a potentially fatal complication of COVID-19. A decreasing level of platelet production and coagulation factors was observed, followed by a consequent increase in their consumption [17]. Thus, SARS-CoV-2 can be secondarily associated with hemorrhagia due to consumption of platelet and coagulation factors [18]. The administration of LMWH (low molecular weight heparin) has been shown to be useful as a treatment for the pregnant population and is ideally administered as soon as possible after the onset of the disease [19–23].

According to Kotlar et al., another pathophysiological mechanism, which is related to the immune mechanism, could explain the occurrence of thrombocytopenia: the occurrence of a cytokine storm and endotheliopathy, which plays a role within the hemostatic process. This may be the reason that it responds so well to glucocorticoid treatment. Life-threatening hemorrhage has been rarely reported in this condition [20,21].

In this study, we aimed to explore the impact of severe acute respiratory syndrome coronavirus-2 (SARS-CoV-2) infection on a pregnant population. This included to what extent this population behaved as a "high-risk category" population, what obstetric consequences infection with this virus had on the mother and fetus, to what extent those devastating changes, which included severe pneumonia, multiorgan failure, thrombosis, hemorrhage, and death, occurred, and, finally, to what extent did the risk factors or demographics described in the literature have an influence on the manifestation of the disease. Moreover, the observed changes in the related hematological or biochemical parameters were of great interest.

## 2. Materials and Methods

### 2.1. Patients

This was a retrospective, single-center study of 78 patients, of which 39 patients who had tested positive for COVID-19 and 39 control subjects were recruited from the Emergency County Hospital of Hunedoara, Romania. The aim of this case series study was to explore the indirect impact of the COVID-19 pandemic on pregnancy. A comprehensive study is essential due to the lack of information and the various consequences of this virus. Our patient group was in the "high-risk" category. The inclusion criteria were defined as follows: pregnant women with a positive RT-PCR result for SARS-CoV-2 infection, who required hospitalization. In that period the active variant of the virus present on Romanian territory was Delta B.1.617.2. The control group consisted of pregnant women admitted for various pathologies or for giving birth, for which the same parameters were followed. All patients with COVID-19 enrolled in this study were diagnosed according to World Health Organization interim guidance [24]. Epidemiological and clinical data were obtained from each inpatient between 1 October 2020 and 31 December 2021. All the procedures in this study were in accordance with the bioethical standards of the Helsinki Declaration and were approved by the Ethics Committee of the Emergency County Hospital of Hune-doara (Nr. 7591/06 June 2022)Statistical analysis was performed using the SPSS statistical package, version 25.0 (IBM Corp., Armonk, NY, USA).

### 2.2. Data Collection

Primary data on the hospitalized patients including age, gestation, parity, socio-economic status, risk factors, laboratory tests, and presenting symptoms were investigated. On laboratory testing, we followed the parameters that reflected changes related to the occurrence of SARS-CoV-2 infection, such as leukocytes (WBCs), neutrophils, lymphocytes, red blood cells (RBCs), hemoglobin (Hgb), hematocrit (Hct), platelets (PLT), international normalized ratio (INR), creatinine, urea, uric acid, aspartate aminotransferase (AST), alanine transaminase (ALT), glucose, C reactive protein, lactate dehydrogenase (LDH), fib-

rinogen, and ferritin. Since the study was conducted on pregnant patients, we had to limit ourselves regarding the use of radiological investigations/computed tomography (CT). Only postpartum patients/those who suffered abortion were investigated in this manner.

The most important parameters were patient age, gestation, parity, and socio-economic status, the last influencing access to a medical doctor, which, in certain situations, was delayed, thus affecting prognosis.

*2.3. Statistical Analysis*

Continuous and normally distributed baseline characteristics are presented as mean ± standard deviation with minimum and maximum. Non-normally distributed variables are presented as median with 25th percentile (Q1) and 75th percentile (Q3). Categorical variables, which are presented as frequency and percentages, were compared between groups with the Chi squared test or Fisher's Exact test when an expected cell percentage was greater than 0.20. Normal distribution was checked with the Shapiro–Wilk test. In the analysis of the difference between the numerical data of the two groups, the independent samples t test was used when the data were distributed in a normal distribution, and the Mann–Whitney U test was used when the data were not normally distributed. Statistical analysis was performed using the SPSS statistical package, version 25.0 (IBM Corp., Armonk, NY, USA). A *p* value < 0.05 was considered statistically significant.

The laboratory tests, which were shown to alter during SARS-CoV-2 virus infection, were monitored to see if they respected the pattern seen in the nonpregnant population. Thus, a statistical analysis was performed from the complete blood count (CBC) to find out to what extent anemia or thrombocytopenia were present in our patients and to what extent lymphocytopenia occurred. Regarding the biochemical analysis, we followed the parameters that underwent changes, especially inflammatory samples, such as polymerase chain reaction (PCR) or lactate dehydrogenase (LDH), with LDH being a marker of cytolysis and tissue damage. Coagulation tests were also carried out due to thrombosis or hemorrhage cases described in the literature.

We have included tables with the most important obstetric complications observed during pregnancy in our patients, such as abortion, preterm birth, preeclampsia (PE), and intrauterine growth restriction (IUGR). Of the included patients, five were still under observation at the time of writing. For those who gave birth, we recorded birth rate, weight, and APGAR score of the newborn, and sex and complications at birth, such as fetal hypoxia, maternal or neonatal death, and vertical transmission of the virus. The newborns were evaluated by real-time postnatal polymerase chain reaction (RT-PCR) viral testing, as were their mothers.

The benefits of the treatments were quantified depending on the evolution of the pathology, the presence of obstetric complications, and the number of days of hospitalization.

**3. Results**

Of the 78 patients who were included in the study, 50% were COVID-19 positive and the rest were COVID-19 negative (control group) according to the RT-PCR testing method. The mean age of the patients was 29.47 ± 6.28 (17–41). In Table 1, the age, gestation, parity, and socio-economic environment status of the patients in the control and COVID-19-positive groups are compared. According to Table 1, there was no statistical difference between the groups in terms of age ($p = 0.421 > 0.05$); there was no statistical difference between the groups in terms of gestation and parity ($p = 0.305 > 0.05$ and $p = 0.208 > 0.05$, respectively); and there was no statistical difference between the groups in terms of the socio-economic environment status of the patients ($p = 0.329 > 0.05$).

**Table 1.** Baseline characteristics of the tested patients: control and COVID-19 positive, in connection with the variables age, gestation, parity.

| | Control (*n* = 39) | | COVID-19 (*n* = 39) | | |
|---|---|---|---|---|---|
| | Mean ± SD (Min.–Max.) | Median (Q1–Q3) | Mean ± SD (Min.–Max.) | Median (Q1–Q3) | *p* |
| Age | 30.05 ± 6.79 (17–41) | | 28.9 ± 5.76 (18–41) | | 0.421 |
| Gestation | 2.77 ± 1.94 (1–9) | 2 (1–3) | 2.18 ± 1.17 (1–6) | 2 (1–3) | 0.305 |
| Parity | 2.15 ± 1.6 (0–6) | 2 (1–3) | 1.67 ± 1.13 (0–6) | 2 (1–2) | 0.208 |

As is evident from Table 2, only neutrophils, AST (aspartate aminotransferase), lactate dehydrogenase (LDH), fibrinogen and ferritin were statistically significant ($p < 0.05$). The lymphocytes value, AST values, and lactate dehydrogenase values of the COVID-19 group patients were higher than those of the control group. The fibrinogen and ferritin values of the positive group were lower than those of the control group. There was no statistical difference between the other blood parameters according to the groups ($p > 0.05$).

**Table 2.** Comparison of the blood parameters according to groups.

| | Normal Range | Control (*n* = 39) | | COVID-19 (*n* = 39) | | |
|---|---|---|---|---|---|---|
| | | Mean ± SD (Min.–Max.) | Median (Q1–Q3) | Mean ± SD (Min.–Max.) | Median (Q1–Q3) | *p* |
| Leukocytes | 4.00–10.00 × 10⁹/L | 12.16 ± 3.26 (7.19–21.9) | 11.98 (9.74–13.81) | 11.34 ± 4.07 (4.08–22.21) | 10.77 (8.67–13.33) | 0.168 |
| Neutrophils | 2–7 × 10⁹/L | 8.91 ± 3 (4.16–18.71) | 8.49 (6.48–10.72) | 8.66 ± 3.63 (2.45–18.88) | 7.9 (6.2–10.11) | 0.442 |
| Lymphocyte | 1.0–4.1 × 10⁹/L | 3.59 ± 3.7 (1.41–17.3) | 2.39 (1.86–3.1) | 1.93 ± 0.83 (0.37–3.91) | 1.96 (1.45–2.38) | 0.002 |
| Red blood cells | 4.7–6.1 million cells/mcL | 4.21 ± 0.48 (3.39–5.38) | 4.16 (3.92–4.54) | 4.24 ± 0.37 (3.42–4.89) | 4.19 (3.94–4.52) | 0.756 |
| Hemoglobin | 11.5–16 g/dL | 11.2 ± 1.18 (8.6–13.5) | 11.5 (10.4–12.1) | 11.86 ± 1.25 (8.4–14.3) | 11.8 (11.2–2.7) | 0.021 |
| Hematocrit | 35–48 % | 36.28 ± 3.71 (28.5–43.4) | 36.5 (33.7–38.5) | 37.58 ± 3.3 (29.2–44.9) | 37.7 (35.5–39.4) | 0.105 |
| Platelets | 150–450 × 10⁹/L | 246.62 ± 75.86 (101–420) | 261 (185–307) | 245.54 ± 83.65 (57–484) | 242 (185–307) | 0.953 |
| International Normalized Ratio | 0.8–1.2 | 0.97 ± 0.12 (0.77–1.24) | 0.99 (0.88–1.07) | 0.99 ± 0.13 (0.8–1.2) | 0.98 (0.88–1.13) | 0.558 |
| Creatinine | 0.50–0.90 mg/dL | 1.9 ± 8.23 (0.37–52) | 0.56 (0.52–0.67) | 2.51 ± 12.08 (0.38–76) | 0.57 (0.5–0.64) | 0.708 |
| Urea | 16–43 mg/dL | 19.51 ± 7 (11.19–48.78) | 17.99 (15.08–22.5) | 18.71 ± 6.16 (8.87–35.62) | 19.24 (13.41–22.3) | 0.881 |
| Uric acid | 2.3–6.10 mg/dL | 3.88 ± 1.1 (2.3–6.69) | 3.67 (3.01–4.62) | 4.2 ± 0.86 (2.94–6.69) | 4.08 (3.55–4.91) | 0.157 |
| AST | 0.00–31.00 U/L | 19.41 ± 7.33 (11–43) | 17 (15–22) | 26.23 ± 15.4 (12–98) | 22 (18–31) | 0.005 |
| ALT | 0.00–34.00 U/L | 15.85 ± 8.74 (4–50) | 14 (11–19) | 21.21 ± 24.65 (6–126) | 13 (10–19) | 0.952 |
| Glucose | 60–115 mg/dL | 89.54 ± 15.77 (66–163) | 87 (80–98) | 91.74 ± 15.31 (71–134) | 92 (80–98) | 0.535 |
| C Reactive Protein | 0.00–5.00 mg/L | 10.34 ± 11.41 (2.07–73.2) | 7.38 (4.93–12.71) | 18.77 ± 31.37 (0.05–150.87) | 8.48 (4.31–15.5) | 0.932 |

**Table 2.** *Cont.*

| | Normal Range | Control (*n* = 39) | | COVID-19 (*n* = 39) | | |
|---|---|---|---|---|---|---|
| | | Mean ± SD (Min.–Max.) | Median (Q1–Q3) | Mean ± SD (Min.–Max.) | Median (Q1–Q3) | *p* |
| Lactate dehydrogenase | 0.00–247.00 U/L | 94.13 ± 79.04 (11–260) | 65 (26–154) | 227.67 ± 82.6 (110.8–418.9) | 216.2 (144.3–262.6) | <0.0001 |
| Fibrinogen | 180–450 mg/dL | 466.19 ± 111.16 (301.6–903.2) | 451 (405.5–484.4) | 396.25 ± 138.09 (79.1–658) | 357.8 (309.5–506.6) | 0.005 |
| Ferritin | 18–160 ng/mL | 78.49 ± 53.9 (11.05–160) | 97 (19–127) | 40.26 ± 67.14 (5.92–424.18) | 22.2 (12.2–49.99) | <0.0001 |

There was no difference between the control and COVID-19 group in terms of antiviral use, anti-inflammatory and analgesics use, and blood products, as shown in Table 3. Anticoagulant, dexamethasone, uterotonic, antibiotic, and vitamin usage were statistically significant when compared between groups. There was a statistical difference between anticoagulant use in the COVID-19-positive patients (94.9%) and the control group (5.1%) (*p* < 0.0001). There was a statistical difference between dexamethasone use in the COVID-19-positive patients (48.7%) and the control group (2.6%) (*p* < 0.0001). There was a statistical difference between uterotonic use in the COVID-19-positive patients (61.5%) and the control group (87.2%) (*p* = 0.01 < 0.05). There was a statistical difference between antibiotic use in the COVID-19-positive patients (89.7%) and the control group (69.2%) (*p* = 0.025 < 0.05). There was a statistical difference between vitamin use in the COVID-19- positive patients (87.2%) and the control group (20.5%) (*p* < 0.0001).

**Table 3.** Comparison of the variables according to COVID 19- positive and control groups.

| | | Control | COVID-19 | Total | *p* |
|---|---|---|---|---|---|
| | | *n* (%) | *n* (%) | *n* (%) | |
| Antiviral | No | 39 (100) | 34 (87.2) | 73 (93.6) | 0.055 |
| | Yes | 0 (0) | 4 (10.3) | 4 (5.1) | |
| | Refused | 0 (0) | 1 (2.6) | 1 (1.3) | |
| | Total | 39 (100) | 39 (100) | 78 (100) | |
| Anticoagulant | No | 37a (94.9) | 2b (5.1) | 39 (50) | <0.0001 |
| | Yes | 2a (5.1) | 37b (94.9) | 39 (50) | |
| | Total | 39 (1) | 39 (1) | 78 (1) | |
| Dexamethasone | No | 38a (97.4) | 20b (51.3) | 58 (74.4) | <0.0001 |
| | Yes | 1a (2.6) | 19b (48.7) | 20 (25.6) | |
| | Total | 39 (100) | 39(100) | 78 (100) | |
| Uterotonic | No | 5a (12.8) | 15b (38.5) | 20 (25.6) | 0.01 |
| | Yes | 34a (87.2) | 24b(61.5) | 58 (74.4) | |
| | Total | 39 (100) | 39 (100) | 78 (100) | |
| Antibiotics | No | 12a (30.8) | 4b (10.3) | 16 (20.5) | 0.025 |
| | Yes | 27a (69.2) | 35b (89.7) | 62 (79.5) | |
| | Total | 39 (100) | 39 (100) | 78 (100) | |
| Anti-inflammatory and Analgesics | No | 5 (12.8) | 4 (10.3) | 9 (11.5) | 0.99 |
| | Yes | 34 (87.2) | 35 (89.7) | 69 (88.5) | |
| | Total | 39 (100) | 39 (100) | 78 (100) | |
| Vitamins | No | 31a (79.5) | 5b (12.8) | 36 (46.2) | <0.0001 |
| | Yes | 8a (20.5) | 34b (87.2) | 42 (53.8) | |
| | Total | 39 (100) | 39 (100) | 78 (100) | |
| Hemostatics | No | 39 (100) | 39 (100) | 78 (100) | |
| | Total | 39 (100) | 39 (100) | 78 (100) | |
| Blood products | No | 39 (100) | 38 (97.4) | 77 (98.7) | 0.99 |
| | Yes | 0 (0) | 1 (2.6) | 1 (1.3) | |
| | Total | 39 (100) | 39 (100) | 78 (100) | |

There was no statistical difference between days of hospitalization between the groups (*p* = 0.056 > 0.05). When the birth weight and APGAR score values of the patients were compared according to the groups, no statistically significant difference were observed (*p* = 0.408 > 0.05 and *p* = 0.197 > 0.05, respectively) as shown in Table 4.

**Table 4.** Comparison between days of hospitalization, birth weight, and APGAR score, between the COVID-19-positive and control groups.

| | Control (*n* = 39) | | COVID-19 (*n* = 39) | | |
|---|---|---|---|---|---|
| | **Mean ± SD (Min.–Max.)** | **Median (Q1–Q3)** | **Mean ± SD (Min.–Max.)** | **Median (Q1–Q3)** | ***p*** |
| Days of hospitalization | 4.18 ± 1.39 (2–9) | 4 (4–5) | 5.03 ± 2.32 (2–13) | 5 (3–7) | 0.056 |
| Birth weight (grams) | 3363.44 ± 520.03 (2550–4450) | 3265 (2925–3700) | 3199.69 ± 477.35 (2180–4200) | 3200 (2900–3450) | 0.408 |
| APGAR score | 7.64 ± 3.38 (0–10) | 9 (8–10) | 7.41 ± 3.24 (0–10) | 9 (8–9) | 0.197 |

As is evident from Table 5, there was no difference between the control and COVID-19 groups according to IUGR, preeclampsia, premature birth, miscarriage, and gender. C-section, vaginal birth and neonatal asphyxia were statistically significant when compared between groups. There was a statistical difference between C-section in COVID-19-positive patients (56.3%) and the control group (25%) (*p* = 0.011 < 0.05).

**Table 5.** Comparison of the variables according to the COVID-19-positive and control groups.

| | | Control | COVID-19 | Total | *p* |
|---|---|---|---|---|---|
| | | *n* (%) | *n* (%) | *n* (%) | |
| IUGR | No | 39 (100) | 37 (94.9) | 76 (97.4) | 0.494 |
| | Yes | 0 (0) | 2 (5.1) | 2 (2.6) | |
| | Total | 39 (100) | 39 (100) | 78 (100) | |
| Preeclampsia | No | 38 (97.4) | 38a (97.4) | 76(97.4) | 0.99 |
| | Yes | 1 (2.6) | 1 (2.6) | 2 (2.6) | |
| | Total | 39 (100) | 39 (100) | 78 (100) | |
| Premature Birth | No | 38 (97.4) | 36 (92.3) | 74 (94.9) | 0.615 |
| | Yes | 1 (2.6) | 3 (7.7) | 4 (5.1) | |
| | Total | 39 (100) | 39 (100) | 78 (100) | |
| Miscarriage | No | 37 (0.949) | 37 (0.949) | 74 (0.949) | 0.99 |
| | Yes | 2 (0.051) | 2 (0.051) | 4 (0.051) | |
| | Total | 39 (100) | 39 (100) | 78 (100) | |
| C-Section | No | 24a (75) | 14b (43.8) | 38 (59.4) | 0.011 |
| | Yes | 8a (25) | 18b (56.3) | 26 (40.6) | |
| | Total | 32 (100) | 32 (100) | 64 (100) | |
| Vaginal Birth | No | 8a (25) | 18b (56.3) | 26 (40.6) | 0.011 |
| | Yes | 24a (75) | 14b (43.8) | 38 (59.4) | |
| | Total | 32 (100) | 32 (100) | 64 (100) | |
| Neonatal Asphyxia | No | 31a (96.9) | 22b (68.8) | 53 (82.8) | 0.003 |
| | Yes | 1a (3.1) | 10b (31.3) | 11 (17.2) | |
| | Total | 32 (100) | 32 (100) | 64 (100) | |
| Gender | Male | 2 (28.6) | 19 (59.4) | 21 (53.8) | 0.215 |
| | Female | 5 (71.4) | 13 (40.6) | 18 (46.2) | |
| | Total | 7 (100) | 32 (100) | 39 (100) | |
| Neonatal COVID-19 | No | 32 (100) | 32 (100) | 64 (100) | |
| | Total | 32 (100) | 32 (100) | 64 (100) | |
| Fetal/Neonatal Death | No | 32 (82.10) | 32 (82.10) | 64 (82.10) | |
| | Total | 32 (82.10) | 32 (82.10) | 64 (82.10) | |

There was a statistical difference between vaginal birth in COVID-19-positive patients (43.8%) and the control group (75%) ($p = 0.011 < 0.05$). There was a statistical difference between neonatal asphyxia in COVID-19-positive patients (31.3%) and the control group (3.1%) ($p = 0.003 < 0.05$).

For the variables investigated as risk factors, univariate analysis and logistic regression analysis were performed, as is shown in Tables 6 and 7.

**Table 6.** Univariate analysis of the variables.

|  |  | Control *n* (%) | COVID-19 *n* (%) | Total *n* (%) | *p* |
|---|---|---|---|---|---|
| Smoking | No | 34(87.2) | 33(84.6) | 67(85.9) | 0.745 |
|  | Yes | 5(12.8) | 6(15.4) | 11(14.1) |  |
|  | Total | 39(100) | 39(100) | 78(100) |  |
| Obesity | No | 35(89.7) | 37(94.9) | 72(92.3) | 0.675 |
|  | Yes | 4(10.3) | 2(5.1) | 6(7.7) |  |
|  | Total | 39(100) | 39(100) | 78(100) |  |
| Comorbidities | No | 37(94.9) | 35(89.7) | 72(92.3) | 0.675 |
|  | Yes | 2(5.1) | 4(10.3) | 6(7.7) |  |
|  | Total | 39(100) | 39(100) | 78(100) |  |
| Diabetes | No | 37(94.9) | 36(92.3) | 73(93.6) | 0.99 |
|  | Yes | 2(5.1) | 3(7.7) | 5(6.4) |  |
|  | Total | 39(100) | 39(100) | 78(100) |  |

Smoking, obesity, comorbidities, and diabetes were factors that did not exhibit any statistical difference depending on whether the patients were COVID-19-positive or not (*p* values are greater than 0.25).

**Table 7.** Logistic regression analysis.

| Variables | B | S.E. | Wald | *p* | Exp(B) | 95% C.I. for EXP(B) | |
|---|---|---|---|---|---|---|---|
|  |  |  |  |  |  | Lower | Upper |
| Smoke (No/Yes) | −0.35 | 0.673 | 0,27 | 0.603 | 0.705 | 0.189 | 2.634 |
| Obesity (No/Yes) | 1.604 | 1.175 | 1.864 | 0.172 | 4.974 | 0.497 | 49.767 |
| Comorbidities (No/Yes) | −22.841 | 40,192.95 | 0 | 0.99 | 0 | 0 | . |
| Diabetes (No/Yes) | 22.071 | 40,192.95 | 0 | 0.99 | $3.85 \times 10^9$ | 0 | . |
| Constant | −0.519 | 1.481 | 0.123 | 0.726 | 0.595 |  |  |

The selected risk factors were not found to be statistically significant (*p* values were greater than 0.05).

Furthermore, in order to investigate which of the patients were most affected and whether there existed an age threshold for the COVID-19-positive and control groups, an ROC analysis was performed (Figure 1).

Our findings demonstrate that the disease is similar for the nonpregnant female population. Severe complications affected a small percentage of patients and comprised one case of severe pneumonia and two cases of coagulation impairment manifested as hemorrhagic shock followed by disseminated intravascular coagulation (DIC) or thrombosis with ulceration, observed during cesarean delivery. This is in contrast to data from the literature in which respiratory failures are more common.

From an obstetrical point of view, cesarean deliveries prevailed, mainly resulting from fetal hypoxia. Newborns were PCR tested, with none exhibiting COVID-19 infection. Thus, we did not observe vertical transmission of the virus in our patients. All complications related to hypoxia were corrected. We had no maternal or neonatal deaths caused by COVID-19.

According to our study, the need for antiviral treatment was limited, but anticoagulants and dexamethasone proved to be effective in terms of improving outcomes.

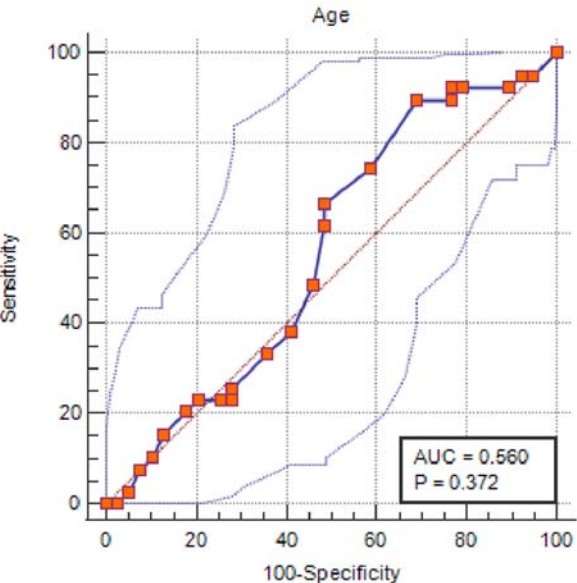

**Figure 1.** The area under curve (AUC) needed to be at least 0.80 *p* value lower than 0.05. However, as we can see in the graph, this was not the situation (0.56 vs. *p* = 0.37).

*Clinical Approaches*

In the following, we would like to share our clinical approaches concerning two cases with increased significance. The first is a case of a patient who developed COVID-19 in the third trimester of pregnancy and underwent cesarean section for obstetric reasons. In addition, in this patient, we intraoperatively discovered a left uterine artery thrombosis that was treated by surgery. It should be noted that this patient was under treatment with a prophylactic dose of low-molecular-weight heparin (LMWH).

The second life-threatening case was that of a 22-year-old patient with a precarious socio-economic status, with a gestational age of 18 weeks, who presented to our department with a massive hemorrhage due to an ongoing late miscarriage. The patient presented asymptomatic SARS-CoV-2, but on laboratory testing, the full blood count (FBC) was notable for lymphopenia 2.05 × 109/L, hemoglobin 11.8 g/dL, hemato-crit 37.7%, severe thrombocytopenia 57 × 109/L, and AST 36 U/L, all of which were shown to have discretely increased. Inflammatory tests indicated minimal changes, CRP 5.86 mg/L and LDH 354.6 U/L, indicating significant cytolysis.

A decreased value of fibrinogen, i.e., 79.1 mg/dL, indicated the consumption of coagulation factors and the occurrence of disseminated intravascular coagulation (DIC) after hospitalization. After admission, the patient suffered a miscarriage. Because of the massive hemorrhage that occurred, our patient required an emergency hysterectomy. The patient was admitted to the intensive care unit (ICU) where she received specific medical care, a transfusion of two units of RBCs and one unit of whole-blood-derived platelets, after which her hemoglobin values (which initially de-creased to 8.3 g/dL) increased to 10.1 g/dL.

Her platelet values increased to 69 × 109/L, and the patient received anticoagulant therapy with low-molecular-weight heparin (LMWH), thus preventing the occurrence of blood clots and the development of disseminated intravascular coagulation (DIC). This was followed by the administration of dexamethasone as a stabilizer for the cell membrane, and to prevent platelet destruction. The patient's condition required an additional platelet transfusion, with values increasing to 105 × 109/L the day after treatment.

For a deficiency of coagulation factors, the patient received a plasma transfusion, with the INR values increasing to 3.42. As her leukocyte levels increased to 14.02 × 109/L and her PCR values increased to 12.79 mg/L, the patient received antibiotic treatment. The follow-up chest CT did not reveal any changes due to SARS-CoV-2 or other secondary

changes. From a surgical and medical point of view, as a result of the rapid and firm intervention and removal of the bleeding factor, the patient's life was saved.

Low-molecular-weight heparin (LMWH) and dexamethasone, once again, proved their effectiveness. We suspect a "cross-section" mechanism between anti-COVID-19 antibodies and platelets. In this way, dexamethasone acts not only as a membrane stabilizer for platelets, but also acts against antibodies developed during SARS-CoV-2 infection. It should be noted, however, that dexamethasone and low-molecular-weight heparin (LMWH) are only effective if therapy is initiated at the onset of disseminated intravascular coagulation (DIC), and these are available and accepted as a therapy during pregnancy, childbirth, or miscarriage. At discharge, the patient was advised to continue the antianemia treatment, prophylactic anticoagulant therapy, to have physical and sexual rest, and to follow a natural hygiene diet. Thereafter, healing was achieved per primam.

Our findings regarding this disease were similar to findings reported for the nonpregnant population. Severe complications affected a small percentage of patients and consisted of one case of severe pneumonia and two cases of manifested coagulation disorder, either by hemorrhagic shock followed by DIC or thrombosis with ulceration found during cesarean delivery. This is in contrast with data from the literature in which respiratory failures are more common [2–4]. Life-threatening hemorrhage has been rarely reported [20,21].

## 4. Discussion

In Romania, the severe acute respiratory syndrome associated with SARS-CoV-2 has become a public health problem. The anatomical and physiological changes seen during pregnancy placed pregnant women in the "high-risk population" for SARS-CoV-2 [1]; however, for the younger patients in our study, the risk of severe complications decreased. We also considered the possible repercussions associated with the vertical transmission of the virus, but RT-PCR detection of the virus in newborns was absent [1,3,5,22–29]. The manifestations of SARS-CoV-2 infection in the pregnant population were similar to those described in the literature for the general population and from other studies on pregnant women [8,24,30]. The mortality rate in our study was zero, whereas the mortality rate from the literature is 3.4%. In our study, the rate of severe cases that required intensive care was much lower than the 50% of SARS-CoV-2-infected pregnant women reported to require intensive care treatment in other studies [10].

Regarding obstetric complications, we experienced an increase in the rate of cesarean delivery and fetal hypoxia corrected after birth, but a decreasing trend regarding other complications described in the literature, such as preeclampsia and HELLP (hemolysis, elevated liver enzymes, and low platelets) syndrome [10]. No substantial intergroup differences were found, including days from admission.

The thrombocytopenia case reported in this study occurred secondary to disseminated intravascular coagulation (DIC) and this complication is not generally seen according to specialized studies [8,10,14,17,21,25,30].

We chose to describe this in detail in order to highlight the importance of low-molecular-weight heparin (LMWH) and dexamethasone treatment as a therapy for SARS-CoV-2 infection in pregnant patients. This treatment was also demonstrated to be effective in thrombotic manifestations in pregnant women, with the related incidence decreasing in our study [13,15,16].

It would be interesting to explore the angiotensin-converting enzyme 2 with the protective effect in acute lung lesions which is inactivated by SARS-CoV-2. ACE2 expression and activity are enhanced during pregnancy and have a role in arterial systolic and diastolic pressure decrease and cardiac output increase. In preeclampsia, like in COVID-19 syndrome, ACE-2 is affected and both preeclampsia and SARS-CoV-2 have a role in endothelial disfunction. [31–33]. We did not have enough patients infected with SARS-CoV-2 who would have developed preeclampsia to justify ACE2 activity.

*Study Limitations and Strong Points*

Given the limitations of our retrospective design, we could not provide a histopathological examination of the placentas after delivery, which is necessary to understand the mechanism by which fetal asphyxia occurred. In our country, the gross and microscopic examination of tissues from COVID-19 patients is prohibited by law.

Another limitation of our study is related to the hyperactivity of the immune system. Although it is part of the pathogenesis of SARS-CoV-2 infection, it is not fully un-derstood yet.

We did not explore the ACE2 activity and how it is influenced by SARS-CoV-2, and how it could induce preeclampsia.

On the other hand, the strong points of this study are related to the fact that we explored thromboembolism as a non-life-threatening complication. Moreover, as a result of pregnancy-related procoagulant factors in pregnant patients, we explored TTP-like changes induced by COVID-19. In addition, we provide recommendations for the use of low-molecular-weight heparin (LMWH) in pregnant patients who test positive for COVID-19, as highlighted in our most severe cases.

We want to recommend exploring solutions regarding maintaining a healthy life-style in the case of a lockdown to reduce the use of anticoagulants. This will have far-reaching economic and social benefits.

## 5. Conclusions

In our study, it was shown that pregnant women do not appear to be more sus-ceptible to SARS-CoV-2 infection than nonpregnant women [23]. However, in those pregnant patients who developed complications, bleeding or thrombosis was observed, probably due to the procoagulant changes related to pregnancy and COVID-19 manifestations, similar to what is seen in thrombotic thrombocytopenic purpura (TTP).

In this study, complications were counteracted in the majority of the situations using a low-molecular-weight heparin (LMWH) treatment as a prompt intervention. This was considered as lifesaving, with the only ambiguity related to this treatment being associated with the dose adjustment criteria.

From an obstetrical point of view, cesarean section prevailed, mainly resulting from fetal hypoxia. The newborns tested negative for COVID-19 by real-time polymerase chain reaction (RT-PCR), so there was no evidence of vertical transmission of SARS-CoV-2 in our patients. All the complications related to hypoxia were corrected. There were no maternal or neonatal deaths during our study.

This new strain of coronavirus, i.e., SARS-CoV-2, has been shown to cause a hetero-geneity of manifestations, such as multiorgan failure, and its evolution remains unknown. In our study, the need for antiviral treatment was limited, but anticoagulants proved effective, thus improving patient outcomes

**Author Contributions:** Conceptualization, M.I.M.; Methodology, I.P. and O.S.C.; Software, D.M.C. and I.E.O.; Validation, M.I.M. and O.S.C.; Formal analysis, I.E.O. and D.M.C.; Investigation, M.I.M., D.M.C. and I.E.O.; Resources, I.P.; Writing—Original draft preparation, M.I.M., D.M.C. and I.E.O.; Writing—Review and editing, M.I.M. and D.M.C.; Visualization, I.P.; Supervision, M.I.M. All authors have read and agreed to the published version of the manuscript.

**Funding:** This research received no external funding.

**Institutional Review Board Statement:** The study was conducted in accordance with the Declaration of Helsinki and approved by the Institutional Ethics Committee of the Emergency County Hospital of Hunedoara (Nr. 7591/6 June 2022).

**Informed Consent Statement:** Informed consent was obtained from all subjects involved in the study.

**Data Availability Statement:** All data produced here are available and can be produced upon request.

**Acknowledgments:** The authors would like to thank all patients and physicians involved in the study with a particular emphasis on our Intensive Care Unit colleagues and healthcare staff. The authors are

also indebted to Havva Serap Toru from Akdeniz University, for support and fruitful discussions, and to Ebru Kaya Başar from the Application and Research Center for Statistical Consultancy, Antalya, Turkey, whose expertise greatly assisted our study.

**Conflicts of Interest:** The authors declare no conflict of interest.

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
