# Peer review of "Evaluation of the Obstetric Patient: Pregnancy Outcomes during COVID-19 Pandemic—A Single-Center Retrospective Study in Romania"

_reports, doi:10.3390/reports5030027_

Round 1
Reviewer 1 Report
Mitranovici and colleagues show a photo of COVID-19 infection in a retrospective single center case series of 39 patients positive for COVID-19 and 39 control subjects recruited from Emergency County Hospital of Hunedoara, Romania. The case series is interesting but in my opinion could be improved with major and minor revisions.
Major revision
-Better explain the conclusion of you study in the asbract
- It would be interesting to understand the period in which this data was collected. If you contextualize the period in which you collect data and you correlate it with the most frequent variant in that period in your country, you should formulate an hypothesis that il not only COV2 related but COV2 variant related, and we know that different variants have different behaviour.
-Go deeper to analyze how pregnancy could be considered a disadvantage or not during SARS-CoV2 infection. What about ACE2 receptors and hormone levels during pregnancy? How is considered preclampsia related to COVID-19 in pregnant positive woman? Have you something data form your patients?
only like examples
- Todros, T., Masturzo, B. & De Francia, S. COVID-19 infection: ACE2, pregnancy and preeclampsia. Eur J Obstet Gynecol Reprod Biol 253, 330, doi:10.1016/j.ejogrb.2020.08.007 (2020).
- Levy, A. et al. ACE2 expression and activity are enhanced during pregnancy. Am J Physiol Regul Integr Comp Physiol 295, R1953-1961, doi:10.1152/ajpregu.90592.2008 (2008).
- Illi, B.; Vasapollo, B.; Valensise, H.; Totta, P. SARS-CoV-2, Endothelial Dysfunction, and the Renin-Angiotensin System (RAS): A Potentially Dangerous Triad for the Development of Pre-Eclampsia. Reprod. Med. 2021, 2, 95-106. https://doi.org/10.3390/reprodmed2020010
-please insert the units of measure in the tables and make the reading clear by aligning the legend to the numbers
Minor revision
-line 18 and line 338 Romania and not our country
-line 22-23 please look the meaning of the sentences and the english form
Thanks a lot for your answer and your job
Author Response
Cover letter 1
Evaluation of the Obstetric patient:Pregnancy Outcomes during COVID-19 Pandemic-A Single center study in Romania
Dear reviewer, thank you very much for your review:
Major review
1).I kept the abstract and I changed the conclusions according to our findings: I added line 404-412: From an obstetrical point of view, cesarean section prevailed, mainly resulting from fetal hypoxia. The newborns tested negative for COVID-19 by real-time polymerase chain reaction (RT-PCR), so there was no evidence of vertical transmission of the SARS-CoV-2 in our patients. All the complications related to hypoxia were cor-rected. There were no maternal or neonatal deaths during our study.
This new strain of coronavirus, i.e., SARS-CoV-2, has been shown to cause a heterogeneity of manifestations, such as multiorgan failure, and its evolution remains unknown. In our study, the need for antiviral treatment was limited, but anticoagulants proved effective, thus improving patient outcomes.
2). In that period between October 2020 and December 2021 the active variant of the virus present on Romanian territory was Delta B.1.617.2.Line 136-137 and 141.
3). These undergo into an increased risk group, due to the anatomical and physiological changes during pregnancy, such as: reduced functional residual volumes, diaphragm elevation, altered cell immunity, increased thoracic transverse diameter, changes in pulmonary volume, vasodilation, mucosal edema are associated with the tendency of mothers to hypoxia. Hormonal changes also play an important role in these vascular and respiratory modifications and also in pregnancy-specific procoagulant events.[1,2].Line 59-64.
Pregnancy being a state of hypercoagulation which is likely an adaptive mechanism to reduce the risk of hemorrhage during and after the delivery process, it presents de-creased fibrinolytic activity and the appearance of venous stasis. In addition, we men-tion prolonged sedentary lifestyle, the consequence of lock-down and limitation of movement due to cesarean section, these also being predisposing factors for thrombo-sis, thromboembolism and thromboembolic risk factors in COVID-19 [13].Line 89-95.
4). It would be interesting to explore the angiotensin-converting enzyme 2 with protective effect in acute lung lesions which is inactivated by SARS-COV-2. ACE2 ex-pression and activity are inhanced during pregnancy and has a role in in arterial sys-tolic and diastolic pressure decrease and cardiac output increase. In preeclampsia like in covid -19 syndrome ACE-2 is affected and both preeclampsia and SARS-COV-2 ha-ve a role in endothelial disfunction.(31,32,33).We did not have enough patients in-fected with SARS-COV-2 wivh would have developed preeclampsia to justify ACE2 activity. Line 364-370. We did not explore the ACE2 activity and how it is influenced by SARS-COV-2and how it could induce preeclampsia.Line 382-383.
5). I introduced the normal values in table 2.
Minor revision
In stad of „our country”I changed with „Romania” line 37 and 339.
I used english editting.
Thank you very much.
Reviewer 2 Report
The topic of the present observational retrospective study, evaluating pregnancy outcomes during COVID-19 pandemic in Romania, may be interesting.
Study design is confused, ranging from retrospective to case-control to case series.
Sample size should be implemented.
Submitted manuscript needs an extensive editing for English language and to partially re-organized. Paragraph 3.1. “Clinical approaches” should be partially moved to Discussion section and described approaches should be compared to those available in the literature to establish evidence-based recommendations in case.
Author Response
Cover letter 2:
Pregnancy Outcomes during COVID-19 Pandemic - A Single-Center Retrospective Study in Romania
Dear reviewer
1). It is a retrospective single center study: line 129.
2).The sample size is composed of 78 patients from which 39 patients positive for COVID-19 and 39 control subjects were recruited from Emergency County Hospital of Hunedoara, Romania. Line 129-130.
3) We moved from Clinical aproaches line 331 to Discussions line 351 and we compared our findings to those available in the literature to establish evidence based recommendations in case. Line 351-363.
Thank you.
Round 2
Reviewer 1 Report
All the comments were made in the manuscript and for this reason I accept it in the present form
Reviewer 2 Report
I congratulate the Authors for the work done